# PERSONALIZED FEDERATED PARTIAL LABEL LEARNING

## ABSTRACT

Partial Label Learning (PLL) is known as a valuable learning technique that trains Machine Learning (ML) models on partial label datasets, where the ground truth label is concealed within the candidate label set of each data instance. It learns label correlation based on a single centralized dataset to predict the latent true label. When data is non-independent and identically distributed (non-i.i.d.) among workers in Federated Learning (FL), the label correlation interference problem occurs. To address the issue, in this paper, we propose pFedPLL, a personalized federated partial label learning algorithm with two new designs. In Label Correlation Isolation (LCI), we first develop a twin-module architecture, where a feature-level correlation matrix layer for each worker is isolated locally to prevent it from being interfered with by others. In Label Correlation Personalization (LCP), we then propose a bi-directional calibration loss to identify a more accurate learning direction, where the positive calibration aligns the prediction result with the latent true label, and the negative calibration pushes away the prediction result that falls into the non-candidate label set. We provide a convergence analysis of pFedPLL with a rate of $O\left(\sqrt{\frac{1}{T}}\right)$ for smooth non-convex problems. Experiment results demonstrate that pFedPLL outperforms SOTA federated PLL algorithms and the federated version of centralized PLL algorithms across nine datasets.

## 1 INTRODUCTION

Federated Learning (FL) (McMahan et al., 2017) has become an emerging topic in recent years due to its advantages in efficient parallel training processes and private data isolation across workers/clients. The performance of FL is highly related to the *quantity*, *quality*, and *heterogeneity* of data. However, due to different data collection environments (non-expert users, limited computational power, and varying geo-locations of edge devices, etc.), it is costly to collect a large amount (high quantity) of high quality (with ground truth label) data instances. One possible solution is to first assign a set of potential true labels to each specific data instance to quickly collect a large quantity of data. Then, the Machine Learning (ML) model learns and analyzes the label correlation to mitigate the negative effect of noisy labels during training. Such a learning method is within the scope of Partial Label Learning (PLL) (Cour et al., 2011), where the candidate label set (containing potential true labels) consists of one ground truth label and several correlated false positive labels (noisy labels). Since the ground truth label is concealed in the candidate label set, PLL aims to train a classifier to predict the latent true label (the most likely true label) for each data instance by analyzing the correlations among the labels within the candidate label set. The model performance is highly dependent on the accurate label correlation learned during training.

The label correlation in PLL has been proven to be a valuable component in centralized ML. It finds a way of training a model with relatively *large quantity* but *low quality* data while maintaining the model performance. However, when data is non-independent and identically distributed (non-i.i.d.) among workers in FL, the centralized label correlation may not work in such *heterogeneous* data scenario. Due to non-i.i.d. data, the label correlation learned from the local dataset is only applicable to each worker. When aggregation occurs, it will cause the *label correlation interference* problem. For example, in worker "A", the image data instance of digit "2" is very similar to digit "3" (high label correlation between digit "2" and "3"), while in worker "B", the digit "2" has a high label correlation with digit "5". During the aggregation phase, the global model aggregates each

worker's label correlation and learns that the digit "2" has a high label correlation with both "3" and "5". When the global model is distributed back to workers, such *label correlation interference* will mislead each worker's unique label correlation information, resulting in the degradation of model performance.

To address this issue, we develop an all-in-one solution, pFedPLL, a personalized federated partial label learning algorithm that efficiently trains FL models on *heterogeneous* partial label data. pFedPLL generally consists of two components: ① *Label Correlation Isolation* (LCI) and ② *Label Correlation Personalization* (LCP).

**LCI.** We develop a new correlation matrix layer that is only updated on the local/worker dataset and is not aggregated to the global model, preventing each worker's unique label correlation information from being interfered with by others. The training layers before the correlation matrix layer are aggregated to obtain the global representative feature information. Please note that we insert the correlation matrix layer in the second-to-last layer of the DNN model instead of the last layer. This is because the last layer learns the exact label correlation (the same dimensions as the number of classes of labels) with coarse granularity. To learn more correlation information, we push the correlation matrix layer to the second-to-last layer by learning feature correlation with fine granularity, where the second-to-last layer usually has more dimensions than the number of classes of labels.

**LCP.** Since the correlation matrix in LCI is isolated locally, the matrix itself is personalized. In order to better utilize the personalized correlation matrix information to help identify a more accurate learning direction, we further develop a **bi-directional calibration** loss. Apart from the basic well-known summarization PLL loss (Feng et al., 2020), we propose *positive calibration* to align the prediction result with the latent true label, and *negative calibration* to push away the prediction result that falls into the non-candidate label set (labels that are not within the candidate label set).

We evaluate our pFedPLL algorithm both theoretically and experimentally. We prove that pFedPLL converges for smooth non-convex problems at a rate of $O\left(\sqrt{\frac{1}{T}}\right)$ over $T$ global iterations. In the experiments, we compare pFedPLL with mainstream federated PLL algorithms and federated version of centralized PLL algorithms across nine datasets. The ablation study of LCI and LCP is also evaluated. The experiment results demonstrate that pFedPLL consistently outperforms all benchmark algorithms with up to $49.93\%$ accuracy increase in various ML settings.

## 2 RELATED WORK

**Partial label learning.** Current approaches to PLL disambiguation can be divided into two categories (Tian et al., 2023): averaging and identification methods. Averaging methods (Cour et al., 2011; Hüllermeier & Beringer, 2005; Zhou & Gu, 2018) treat all candidate labels equally as ground truth. Further advancing this, Gong et al. (2021) propose a discriminative metric learning approach using a Mahalanobis distance metric to assess similarity between neighbors with similar labels. However, such predictions are easily misled by false positive labels. On the other hand, identification methods (Liu & Dietterich, 2012) aim to discover latent true labels during training, using techniques such as Expectation Maximization (EM) algorithms (Jin & Ghahramani, 2002) and maximum margin methods (Nguyen & Caruana, 2008; Yu & Zhang, 2016) to resolve ambiguity. However, Both approaches, however, rely on global information, such as label distribution in EM and distance metrics, which conflict with privacy requirements in federated learning (FL). As a result, traditional PLL methods encounter significant challenges when applied directly in FL environments. Recently, DL-based methodologies(Wen et al., 2021; Feng et al., 2020) have emerged to address the PLL problem. Zhang et al. (2021) introduced Class Activation Value (CAV) to transform PLL into a supervised learning problem. Although these methods perform well in centralized settings, simply migrating them to an FL environment may still not work. Our proposed pFedPLL algorithm provides a series of methods (LCI & LCP) to enable training models in federated partial label datasets.

**Federated learning.** FL (McMahan et al., 2017) allows local models to collaboratively train a global model while keeping data private. However, the non-i.i.d. nature of local data can degrade global model performance (Zhu et al., 2021). To address this, Personalized Federated Learning (PFL) (Tan et al., 2022) was developed to adapt the global model to individual workers. Methods like Split Learning (SL) (Vepakomma et al., 2018) and the "base layers + personalized layers" design (Arivazhagan et al., 2019) help manage data heterogeneity by decoupling shared and local

model components. However, these approaches have mostly been tested on supervised datasets and have yet to be adapted for PLL scenarios. Our pFedPLL algorithm personalizes part of the worker's model (LCP) to address the issue of non-i.i.d. partial label datasets.

**Weakly supervised learning in FL.** Weakly supervised learning includes Semi-supervised Learning (SSL), Noise Label Learning (NLL), Multi-instance Learning (MIL), and PLL, but studies on PLL in FL are limited. Recently, Yan & Guo (2024) introduced FedPLL_LAAR, which reduces client drift via adaptive gradient alignment regularization but neglects label correlations. The method assumes a class-dependent generation process (Lv et al., 2020), but in realistic scenarios, partial label datasets should follow an instance-dependent generation process (Xu et al., 2021), where feature-related false labels are more likely to enter the candidate label set, leading to stronger label correlations. Besides PLL, other weakly supervised learning methods like SSL, NLL, and MIL are well-studied in FL. Unlike PLL, these methods deal with single or unlabeled data and don't involve selecting the correct label from a candidate set. For example, FedMatch (Jeong et al., 2020) in SSL uses labeled data as anchors to improve performance, while PLL faces greater challenges due to uncertain true labels. NLL in FL, studied by (Song et al., 2022), focuses on mitigating the effect of noisy data (incorrectly labeled), while PLL must identify the correct one from a set. FedMIL (Bastola et al., 2024) focuses on bag-level predictions in MIL rather than identifying the latent true label. By developing a feature-level correlation matrix layer in LCI, pFedPLL utilizes knowledge from label correlations to effectively identify the latent true label from the candidate label set.

Considering the growing concerns about data privacy and the high costs of data labeling, exploring PLL in an FL environment is essential. This paper investigates the PLL problem in FL, synergistically utilizing the advantages of DL-based PLL methods and FL.

## 3 METHOD: PFEDPLL

### 3.1 PROBLEM FORMULATION

We consider a typical FL system consisting of $K$ workers/clients (indexed by $k$) and one aggregator. The worker $k$ maintains a local partial label dataset $D_k \triangleq \{(\boldsymbol{x}_i^k, Y_i^k)\}_{i=1}^{|D_k|}$, where $|D_k|$ is the total number of data samples in $D_k$, $\boldsymbol{x}_i^k$ is the $i$th data instance in worker $k$, and $Y_i^k \in \{0,1\}^C$ is the label set of $\boldsymbol{x}_i^k$ across $C$ label classes. Let $M_i^k = \{j \in C \mid y_{i,j}^k = 1, y_{i,j}^k \in Y_i^k\}$ denote the candidate label set and $\bar{M}_i^k = \{j \in C \mid y_{i,j}^k = 0, y_{i,j}^k \in Y_i^k\}$ denote the non-candidate label set. The ground truth label is known to reside in the corresponding candidate label set, i.e., $y_{i,j}^k \in Y_i^k, \exists j \in M_i^k$, but cannot be directly accessible, bringing significant challenges for training. The objective is to find the optimal model $\boldsymbol{w}^*$ that minimizes the global loss function

$$\min_{\boldsymbol{w} \in \mathbb{R}^d} F(\boldsymbol{w}) \triangleq \sum_{k=1}^{K} F_k(\boldsymbol{w}, D_k), \tag{1}$$

where $d$ is the dimension of model $\boldsymbol{w}$, $F(\boldsymbol{w})$ is the global loss function, and $F_k(\boldsymbol{w})$ is the worker $k$'s loss function. Here, we define the global loss function $F(\boldsymbol{w})$ as the sum of all workers' loss functions $F_k(\boldsymbol{w})$, because the weight/contribution of each worker loss function is learnt during the training, rather than being predefined.

### 3.2 LCI: LABEL CORRELATION ISOLATION

The typical label correlation mechanism works well in centralized PLL (Xu et al., 2020). It learns the label correlation from a single centralized dataset. When training FL models on decentralized heterogeneous (non-i.i.d.) datasets, each worker can only learn its own label correlation from its own dataset. When aggregation occurs, one worker's label correlation might mislead others due to non-i.i.d. data. To this end, we develop the Label Correlation Isolation (LCI) mechanism, which keeps the label correlation information local and thus prevents it from being interfered with by others.

We start by considering a general Deep Neural Network (DNN) with $n$ layers. We first propose a label correlation matrix layer with the dimension of $p \times p$. Then, we insert the layer into the second-to-last position of the original DNN to form a new $(n + 1)$ layers DNN, where the label correlation matrix is the $n$th layer. Here, $p$ varies depending on the architecture of the DNN model

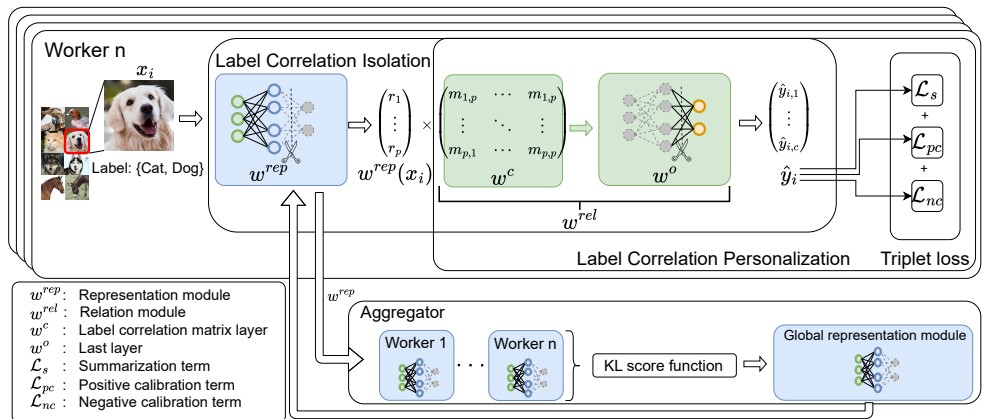

Figure 1: An overview of pFedPLL. Each data instance in the partial label dataset is linked to a candidate label set, with only one true label that remains unknown during training. Label Correlation Isolation (LCI) is achieved through the **twin-module** architecture ($\boldsymbol{w}_{rep}\&\boldsymbol{w}_{rel}$), where only the representation module is aggregated in the server. Label Correlation Personalization (LCP) is accomplished by training the model with the triplet loss including **bi-directional calibration** ($\mathcal{L}_{pc}\&\mathcal{L}_{nc}$).

but is larger than the number of label classes ($p > C$). Therefore, the matrix contains more fine-grained information for better performance (feature-level correlation vs. label-level correlation). Let $\boldsymbol{w}^{rep}, \boldsymbol{w}^c$, and $\boldsymbol{w}^o$ denote the 1st to $(n-1)$th layers, $n$th layer, and $(n+1)$th layer respectively. The label prediction $\hat{\boldsymbol{y}}_i$ for $\boldsymbol{x}_i$ is calculated as

$$\hat{\boldsymbol{y}}_i = \boldsymbol{w}^o(\boldsymbol{w}^c(\boldsymbol{w}^{rep}(\boldsymbol{x}_i)). \tag{2}$$

Since the correlation matrix is isolated within each worker, we introduce a **twin-module** architecture where $\boldsymbol{w}^{rep}$ is the representation module and $\boldsymbol{w}^{rel} \triangleq [\boldsymbol{w}^c, \boldsymbol{w}^o]$ denotes the relation module. $\boldsymbol{w}^{rep}$ is aggregated and redistributed back to each worker to obtain robust/global feature representation information. $\boldsymbol{w}^{rel}$ contains the worker's unique feature-level correlation and is kept locally (not being aggregated). $\boldsymbol{w}^c$ is initialized as a $p \times p$ diagonal matrix with "1" on the diagonal and "0" elsewhere when training begins. By adding superscript $k$ (worker index), and subscripts $i$ (data instance index), $t$ (global iteration index), and $h$ (local iteration index), the prediction of the $i$th data instance for $k$th worker at the $h$th local iteration in the $t$th global iteration is calculated as

$$\hat{\boldsymbol{y}}_{i,t,h}^k = \boldsymbol{w}_{t,h}^{k,rel}(\boldsymbol{w}_{t,h}^{k,rep}(\boldsymbol{x}_i)), \forall i \in |D_k|. \tag{3}$$

The complete worker $k$'s model is then denoted by $\boldsymbol{w}_{t,h}^k \triangleq [\boldsymbol{w}_{t,h}^{k,rep}, \boldsymbol{w}_{t,h}^{k,rel}]$.

In summary, by implementing the twin-module architecture, we address the label correlation interference issue while maintaining the robust model performance.

### 3.3 LCP: Label Correlation Personalization

We have utilized LCI in Section 3.2 to prevent each worker's correlation information from being interfered with by others. In this circumstance, the worker's unique correlation matrix is personalized. The next step is to further utilize this information to help identify a more accurate learning direction for each worker. On the basis of the summarization PLL loss (Feng et al., 2020), we propose **bi-directional calibration** loss for each worker to encourage the correct prediction (aligning with the latent true label) while discouraging the false prediction (prediction falls into the non-candidate label set).

**Summarization.** We begin by introducing the summarization loss, a well-known loss used in PLL which lets the model's prediction fall into the candidate label set. For convenient presentation, we omit iteration indexes and worker index ($t, h$, and $k$) for now. Let $\tilde{\boldsymbol{y}}_i$ denote the prediction (distribution) of the model after normalization (softmax) of $\hat{\boldsymbol{y}}_i$, i.e.,

$$\tilde{y}_{i,j} = \texttt{softmax}(\hat{y}_{i,j}) = \frac{\exp(\hat{y}_{i,j})}{\sum_{c=1}^C \exp(\hat{y}_{i,c})}, \forall \hat{y}_{i,c} \in \hat{\boldsymbol{y}}_i, \tilde{y}_{i,j} \in \tilde{\boldsymbol{y}}_i. \tag{4}$$

Then, we define the summarization loss as

$$\mathcal{L}_s = -\log\left(\sum_{j \in M_i} \tilde{y}_{i,j}\right). \tag{5}$$

It calculates the predicted values of all labels within the candidate label set, but treats them equally (simple summation without weight). Therefore, by minimizing $\mathcal{L}_s$, we can only foster the prediction result falling into the candidate label set.

**Positive calibration.** $\mathcal{L}_s$ has made the prediction fall into the candidate label set. Then, we zoom in on the candidate label set to distinguish each candidate label so as to find the latent true label. Let $\boldsymbol{\alpha}_i = [\alpha_{i,1}, \alpha_{i,2}, \ldots, \alpha_{i,C}]^\top$ denote the confidence of each label for data instance $\boldsymbol{x}_i$, where $\sum_{j \in M_i} \alpha_{i,j} = 1, \alpha_{i,j} \geq 0, \forall j \in M_i$, and $\alpha_{i,j} = 0, \forall j \in \bar{M}_i$. Please note $\boldsymbol{\alpha}_i$ dynamically evolves during the training, and we only use the confidence related to candidate labels. When training begins, $\alpha_{i,j}$ is initialized as $\frac{1}{|M_i|}, \forall j \in M_i$, where $|M_i|$ is the size of $M_i$. During training, $\alpha_{i,j}$ is updated as

$$\alpha_{i,j} = \frac{\tilde{y}_{i,j}}{\sum_{m \in M_i} \tilde{y}_{i,m}}, \forall j \in M_i. \tag{6}$$

Then, we define the positive calibration loss as

$$\mathcal{L}_{pc} = -\sum_{j \in M_i} \alpha_{i,j} \log(\tilde{y}_{i,j}). \tag{7}$$

Here, we put the confidence $\alpha_{i,j}$ as the weight/factor of $\log(\tilde{y}_{i,j})$, which aligns the confidence with the predicted result for a specific label $j$ in data instance $\boldsymbol{x}_i$. Therefore, when both $\alpha_{i,j}$ and $\tilde{y}_{i,j}$ are large, the $\mathcal{L}_{pc}$ is minimized, implying the nature of positive calibration, where the predicted label with high weight (latent true label) should have high confidence. In the meanwhile, the noisy information from potential false-positive labels is alleviated.

**Negative calibration.** When we zoom out on the whole label set (candidate label set vs. non-candidate label set), any prediction that falls into the non-candidate label set is wrong. To this end, we develop a negative calibration that pushes the prediction away from the non-candidate label set. We select the label with the highest weight in the non-candidate label set and give a penalty to this label (adding a "minus"). The negative calibration loss is calculated as

$$\mathcal{L}_{nc} = -\log(1 - \max\{\tilde{y}_{i,j} \in \tilde{\boldsymbol{y}}_i \mid j \in \bar{M}_i\}). \tag{8}$$

By minimizing $\mathcal{L}_{nc}$, the probability of the highest label prediction in non-candidate label set is minimized, implying the more likely prediction in the candidate label set.

**Triplet loss.** To summarize, we construct the final triplet loss for each worker $k$ as follows (iteration indexes and worker index ($t, h$, and $k$) are omitted on the right hand side):

$$F_k(\cdot) = \lambda_1 \mathcal{L}_s + \lambda_2 \mathcal{L}_{pc} + \lambda_3 \mathcal{L}_{nc}, \tag{9}$$

where $\lambda_1$, $\lambda_2$, and $\lambda_3$ are hyperparameters that can be adjusted before the training begins.

In summary, by implementing bi-directional calibration, we better utilize the personalized worker correlation information to predict the latent true label for better model performance.

### 3.4 IMPLEMENTATION

The pFedPLL is implemented in Algorithm 1 with $T$ total global iterations (indexed by $t$) and $\tau$ local iterations (indexed by $h \in [0, \ldots, \tau - 1]$) between two consecutive global iterations. As shown in Fig. 1, the pFedPLL generally comprises worker update with triplet loss calculations for LCP, and aggregator update with LCI and KL score calculation for dynamic model weight/contribution.

**Worker update.** At each local iteration $h \in [0, \ldots, \tau - 1]$ for the $t$th global iteration, each worker first randomly fetches a mini-batch $\xi$ from $D_k$ (Line 6) and calculates the worker model loss (Line 7). Then, each worker updates its representation module (Line 8) and relation module (Line 9) via related gradient descent with $\nabla \boldsymbol{w}^{rep} F_k(\boldsymbol{w}_{t,h}^k, \xi)$ for representation module and $\nabla \boldsymbol{w}^{rel} F_k(\boldsymbol{w}_{t,h}^k, \xi)$ for relation module. $\eta$ is the learning rate. Afterward, the confidence of each data instance in $\xi$ is

---

**Algorithm 1** pFedPLL algorithm

---

**Input**: $\tau, T, \eta, K$
**Output**: Final global representation module $\boldsymbol{w}_T^{rep}$

1: For each worker $k$, initialize: $\boldsymbol{w}_{0,0}^{k,rep}, \boldsymbol{w}_{0,0}^{k,rel}, \forall k \in K$, as the same value respectively,
  and $\alpha_{i,j}^k = \frac{1}{|M_i^k|}, \forall j \in M_i^k, k \in K$, where $|M_i^k|$ is the size of $M_i^k$.
2: For the aggregator, initialize: $\boldsymbol{w}_0^{rep} = \boldsymbol{w}_{0,0}^{k,rep}$.
3: **for** $t = 0, 1, \ldots, T - 1$ **do**
4:   For each worker $k = 1, \ldots, K$ in parallel:
5:   **for** $h = 0, 1, \ldots, \tau - 1$ **do**
6:     Randomly fetch mini-batch $\xi$ from $D_k$.
7:     Calculate triplet loss $F_k(\boldsymbol{w}_{t,h}^k, \xi)$ as equation 9.
8:     $\boldsymbol{w}_{t,h+1}^{k,rep} = \boldsymbol{w}_{t,h}^{k,rep} - \eta \nabla_{\boldsymbol{w}^{rep}} F_k(\boldsymbol{w}_{t,h}^k, \xi)$
      //Update worker $k$'s representation module
9:     $\boldsymbol{w}_{t,h+1}^{k,rel} = \boldsymbol{w}_{t,h}^{k,rel} - \eta \nabla_{\boldsymbol{w}^{rel}} F_k(\boldsymbol{w}_{t,h}^k, \xi)$
      //Update worker $k$'s relation module
10:    For each instance $i$ in mini-batch $\xi$, update confidence
11:     $\alpha_{i,j} = \frac{\tilde{y}_{i,j}}{\sum_{m \in M_i} \tilde{y}_{i,m}}, \forall j \in M_i$.
12:    **if** $h == \tau - 1$ **then**
13:     Send $\boldsymbol{w}_{t,\tau-1}^{k,rep}$ to the aggregator.
14:    **end if**
15:   **end for**
16:   For the aggregator:
17:   $\mathcal{S}_t = \text{KL\_Score}\,(w_t^{rep}, w_{t,\tau-1}^{1,rep}, \ldots, w_{t,\tau-1}^{k,rep}, K, B)$
    //Call KL_Score function in Algorithm 2
18:   $\boldsymbol{w}_{t+1}^{rep} = \sum_{k=1}^K s_t^k \boldsymbol{w}_{t,\tau-1}^{k,rep}, \forall s_t^k \in \mathcal{S}_t$
    //Aggregate global representation module
19:   $\boldsymbol{w}_{t+1,0}^{k,rep} = \boldsymbol{w}_{t+1}^{rep}$ for each worker $k$.
    //Distribute global representation module to workers
20:   For each worker $k = 1, \ldots, K$ in parallel:
21:   $\boldsymbol{w}_{t+1,0}^{k,rel} = \boldsymbol{w}_{t,\tau-1}^{k,rel}$
    //Retain worker's own relation module
22:   $\boldsymbol{w}_{t+1,0}^k = [\boldsymbol{w}_{t+1,0}^{k,rep}, \boldsymbol{w}_{t+1,0}^{k,rel}]$
    //Construct worker's complete model
23: **end for**

---

**Algorithm 2** KL_Score function

---

**Input**: $w_t^{rep}, w_{t,\tau-1}^{1,rep}, \ldots, w_{t,\tau-1}^{k,rep}, K, B$
**Output**: $\mathcal{S}_t = \{s_t^1, s_t^2, \ldots, s_t^K\}$

1: For each batch $b \in B$ and worker $k \in K$:
2: Calculate KL divergence $\mathcal{L}_{k,b}^t = \text{KL}(\boldsymbol{w}_{t,\tau-1}^{k,rep}(a_b) || \boldsymbol{w}_t^{rep}(a_b))$.
3: For each worker $k \in K$:
4: Obtain the averaged distance $\ell_t^k = \frac{1}{B} \sum_{b=1}^B \mathcal{L}_{k,b}^t$.
5: Obtain the score $s_t^k = \frac{\ell_t^k}{\sum_{i=1}^K (\ell_t^i)}$.

---

updated (Line 11). Finally, when $h == \tau - 1$ (the end of local iteration for the $t$th global iteration), each worker sends its representation module to the aggregator (Line 13).

**Aggregator update.** At each global iteration $t \in [0, \ldots, T-1]$, when $h == \tau - 1$ (the end of the local iteration for the $t$th global iteration), the aggregator first calls the KL_Score function (Algorithm 2) to calculate scores for all workers (Line 17). Then, the aggregator aggregates the global representation module using the scores as weights (Line 18) and distributes it back to each

worker (Line 19). Each worker's unique relation module is kept locally (Line 21), and the updated complete worker model is constructed for the next round of local iteration (Line 22).

**KL score.** To better measure the weight/contribution of worker models towards the global model, we develop a dynamic weight assignment algorithm in Algorithm 2, using the Kullback–Leibler (KL) divergence (Press et al., 2007) to measure the distance/similarity between the global representation module $\boldsymbol{w}_t^{rep}$ and each worker $k$'s representation module $\boldsymbol{w}_{t,\tau-1}^{k,rep}$. We first feed the same $B$ batches of data $\mathcal{A} = \{a_1, \ldots, a_B\}$ (randomly selected from the validation dataset[1]) to $\boldsymbol{w}_t^{rep}$ and $\boldsymbol{w}_{t,\tau-1}^{k,rep}$, and then compare the predictions using the KL divergence (Line 2 in Algorithm 2),

$$\mathcal{L}_{k,b}^t = \mathtt{KL}(\boldsymbol{w}_{t,\tau-1}^{k,rep}(a_b)||\boldsymbol{w}_t^{rep}(a_b)), \forall b \in B, \tag{10}$$

where $a_b$ is the $b$th batch in $\mathcal{A}$. We repeat the process for each batch and each worker to obtain the distance matrix $\mathbf{L}^t = \left[\mathcal{L}_{k,b}^t\right]_{K \times B}$. Finally, we average the values in each row to obtain the averaged distance for each worker $\ell_t^k = \frac{1}{B}\sum_{b=1}^B \mathcal{L}_{k,b}^t$ (Line 4), which is then normalized to obtain the final score $s_t^k = \frac{\ell_t^k}{\sum_{i=1}^K (\ell_t^i)}, \forall k \in K$ (Line 5). A higher score means a longer distance between the worker and global representation modules, suggesting that such worker representation module should contribute more to the global representation module, and thus we assign such score as the weight (Line 18 in Algorithm 1). The output for Algorithm 2 are scores for all workers in the $t$th global iteration $\mathcal{S}_t = \{s_t^1, \ldots, s_t^K\}$, where $\sum_{k \in K} s_t^k = 1$.

## 4 CONVERGENCE ANALYSIS

To prove the convergence, we propose a virtual relation module update as if each worker's isolated relation module is aggregated and redistributed, i.e., $\boldsymbol{w}_{t+1}^{rel} = \sum_{k=1}^K s_t^k \boldsymbol{w}_{t,\tau-1}^{k,rel}$ and $\boldsymbol{w}_{t+1,0}^{k,rel} = \boldsymbol{w}_{t+1}^{rel}$. Then, the complete virtual global model is denoted by $\boldsymbol{w}_{t+1} \triangleq [\boldsymbol{w}_{t+1}^{rep}, \boldsymbol{w}_{t+1}^{rel}]$. According to the update rules (Lines 8–9 and 18–19) in Algorithm 1, after performing the mathematical transformations, we derive

$$\boldsymbol{w}_{t+1} = \boldsymbol{w}_t - \eta \sum_{k=1}^K s_t^k \sum_{h=0}^{\tau-1} [\nabla_{\boldsymbol{w}^{rep}} F_k(\boldsymbol{w}_{t,h}^k), \nabla_{\boldsymbol{w}^{rel}} F_k(\boldsymbol{w}_{t,h}^k)], \tag{11}$$

which is the basis to prove the convergence. We assume that the gradient of $F_k(\cdot)$ and $F(\cdot)$ is $L$-Lipschitz and has bounded diversity, i.e., $\|\nabla F(\boldsymbol{w}_1) - \nabla F(\boldsymbol{w}_2)\|_2 \leq L\|\boldsymbol{w}_1 - \boldsymbol{w}_2\|_2$, $\|\nabla F_k(\boldsymbol{w}_1) - \nabla F_k(\boldsymbol{w}_2)\|_2 \leq L\|\boldsymbol{w}_1 - \boldsymbol{w}_2\|_2$, and $\mathbb{E}_{\xi \sim D_k}\|\nabla F_k(\boldsymbol{w}_1, \xi) - \nabla F_k(\boldsymbol{w}_1)\|_2^2 \leq \delta^2, \forall \boldsymbol{w}_1, \boldsymbol{w}_2, k$, which are necessary conditions for convergence analysis in the literature (Wang et al., 2019; Yang et al., 2022; Huo et al., 2020).

**Theorem 1.** *Suppose (1) $\eta \leq \frac{1}{2L\tau}$, and (2) $\exists F_{inf}$ is the lower bound of $F(\cdot)$, we have*

$$\min_{t \in [0,\ldots,T-1]} \mathbb{E}_{\xi,k}\|\nabla F(\boldsymbol{w}_t)\|_2^2 \leq \frac{4L}{T}(F(\boldsymbol{w}_0) - F_{inf})) + 3\eta^2 L^2 \delta^2. \tag{12}$$

*Proof.* See complete proof in Appendix A. $\square$

Theorem 1 demonstrates that the square of the global model gradient is upper bounded by a function that is inversely proportional to $T$. The output of the Algorithm 1, $\boldsymbol{w}^{rep}$ is included in the global model $\boldsymbol{w}$, i.e., $\boldsymbol{w} = [\boldsymbol{w}^{rep}, \boldsymbol{w}^{rel}]$. Therefore, we prove that the pFedPLL is convergent with the convergence rate of $O\left(\sqrt{\frac{1}{T}}\right)$ for smooth non-convex problems.

## 5 EXPERIMENTS

### 5.1 EXPERIMENT SETUP

**Comparison methods.** We compare pFedPLL with the federated PLL algorithm (FedPLL_LAAR), federated version of centralized PLL algorithms (Fed_CC, Fed_RC, Fed_CVAL, and Fed_LW), and

---

[1]The validation dataset is randomly selected, comprising 20% of the test dataset, and is not used for training.

the classic FedAvg (McMahan et al., 2017). For FedPLL_LAAR (Yan & Guo, 2024), most hyperparameter settings are from the original paper, but others are fine-tuned to improve performance in our settings. For the federated version of centralized PLL algorithms, Fed_CC, Fed_RC, Fed_CVAL, and Fed_LW are from the classifier-consistent approach (Feng et al., 2020), the risk-consistent approach (Feng et al., 2020), CVAL (Zhang et al., 2021), and Leverage Weighted loss (Wen et al., 2021) respectively.

**Datasets.** We utilize four benchmark datasets (MNIST (LeCun et al., 1998), Fashion-MNIST (F-MNIST) (Xiao et al., 2017), Kuzushiji-MNIST (K-MNIST) (Clanuwat et al., 2018), and CIFAR-10 (Krizhevsky et al., 2009)) and five real-world partial label datasets (Lost (Cour et al., 2011), BirdSong (Briggs et al., 2012), MSRCv2 (Liu & Dietterich, 2012), Soccer Player (Zeng et al., 2013), and Yahoo!News (Guillaumin et al., 2010)). Please note that the benchmark datasets are originally intended for supervised learning, and we manually convert them into partial label datasets. Details of these datasets can be found in Appendix B.1.

**Data generation.** To convert benchmark datasets into partial label datasets, we follow the instance-dependent generation process (Xu et al., 2021), where the flipping probability of each incorrect label is determined by the confidence prediction of a clean neural network trained on a supervised dataset, and $\rho \in [0, 1]$ is the temperature hyperparameter used to control the size of the candidate label set. Please refer to Appendix B.2 for the detailed process. For non-i.i.d. data generation, we use a Dirichlet distribution $Dir(\beta)$ (Minka, 2000) to generate local data for each worker, where $\beta \in (0, +\infty)$ is a hyperparameter that controls the level of data heterogeneity. A smaller $\beta$ indicates a higher level of non-i.i.d. In all experiments, we set $\beta = 0.5$. For accuracy assessment, since the aggregator lacks a complete model (only a global representation module exists), we assess the final accuracy as the averaged accuracy from all workers. Following the same methods (Tan et al., 2023; Lu et al., 2022), we split each worker's dataset into training (80%) and testing (20%) datasets without any data instance overlap and then test each worker's accuracy to obtain the final accuracy.

**Equipment and hyperparameter settings.** The experiments are carried out on a GPU tower server equipped with 4 NVIDIA GeForce RTX 3090 GPUs. In all experiments, models are updated by mini-batch SGD with learning rate $\eta = 0.01$, momentum factor $= 0.9$, and $\lambda_1 = \lambda_2 = \lambda_3 = 1$. Other hyperparameter settings are specified in Appendix B.3 Table 4.

## 5.2 MAIN EXPERIMENT RESULT

Table 1: Accuracy (%) comparisons on benchmark and real-world partial label datasets when $T = 100$. We use a 5-layer LeNet (LeCun et al., 1998) for MNIST, K-MNIST, and F-MNIST, a 34-layer ResNet (He et al., 2016) for CIFAR-10, and a 2-layer MLP for real-world datasets.

| | | pFedPLL | Fed_LW | Fed_CC | Fed_RC | Fed_CVAL | FedPLL_LAAR | FedAvg |
|---|---|---|---|---|---|---|---|---|
| Benchmark dataset | MNIST | **98.14** ± 0.06 | 97.04 ± 0.05 | 96.60 ± 0.08 | 96.44 ± 0.03 | 48.21 ± 0.06 | 82.01 ± 0.03 | 92.69 ± 0.04 |
| | K-MNIST | **89.15** ± 0.03 | 81.77 ± 0.08 | 79.87 ± 0.09 | 78.81 ± 0.08 | 47.79 ± 0.08 | 68.31 ± 0.04 | 73.82 ± 0.05 |
| | F-MNIST | **84.06** ± 0.07 | 82.89 ± 0.11 | 82.35 ± 0.13 | 80.17 ± 0.09 | 46.35 ± 0.15 | 51.23 ± 0.05 | 80.12 ± 0.09 |
| | CIFAR-10 | **82.10** ± 0.11 | 67.94 ± 0.07 | 70.91 ± 0.09 | 59.56 ± 0.14 | 45.96 ± 0.11 | 48.10 ± 0.13 | 62.10 ± 0.19 |
| Real-world partial label dataset | Lost | **56.04** ± 0.04 | 55.29 ± 0.13 | 55.03 ± 0.06 | 53.87 ± 0.04 | 38.82 ± 0.07 | 52.96 ± 0.05 | 47.69 ± 0.03 |
| | Birdsong | **77.94** ± 0.05 | 72.71 ± 0.09 | 71.32 ± 0.05 | 72.19 ± 0.05 | 63.76 ± 0.08 | 66.78 ± 0.07 | 65.59 ± 0.05 |
| | MSRCv2 | **56.10** ± 0.03 | 49.98 ± 0.07 | 49.16 ± 0.04 | 50.19 ± 0.07 | 35.43 ± 0.07 | 46.31 ± 0.05 | 41.60 ± 0.08 |
| | Yahoo!News | **62.35** ± 0.07 | 52.48 ± 0.09 | 52.10 ± 0.05 | 52.16 ± 0.11 | 44.13 ± 0.11 | 50.98 ± 0.13 | 49.35 ± 0.17 |
| | SoccerPlayer | **40.68** ± 0.09 | 37.58 ± 0.06 | 37.40 ± 0.12 | 37.65 ± 0.09 | 37.16 ± 0.10 | 39.17 ± 0.11 | 40.31 ± 0.12 |

**Benchmark datasets.** We evaluate our pFedPLL algorithm using four benchmark datasets: MNIST, F-MNIST, K-MNIST, and CIFAR-10, all adapted to partial label datasets. Table 1 demonstrates that pFedPLL outperforms all benchmarks, with a 1.1% accuracy improvement on MNIST, 7.38% on K-MNIST, and 1.17% on F-MNIST compared to Fed_LW, the second-best algorithm. On CIFAR-10, pFedPLL exceeds Fed_CC, the second-best algorithm, by 11.19%. Federated version of centralized PLL methods (Fed_LW, Fed_CC, Fed_RC, Fed_CVAL) lack mechanisms to mitigate the non-i.i.d. issue in FL, leading to suboptimal performance. In contrast, pFedPLL's LCI design effectively addresses this issue by preventing interference from other workers' models. We also observe that FedPLL_LAAR does not perform well. This is because it utilizes a class-dependent generation process, where only labels with large differences are included in the candidate label set (e.g., horse vs. cat). Nevertheless, in our experiment, we implement an instance-dependent generation process, where similar labels are included in the candidate label set (e.g., horse vs. donkey), making disambiguation more difficult. In pFedPLL, we implement a fine-grained feature-level correlation matrix

and bi-directional calibration loss to distinguish similar labels, leading to superior performance. Overall, pFedPLL consistently outperforms all benchmarks, with training accuracy improvements ranging from 1.1–49.93%.

**Real-world partial label datasets.** We evaluate our pFedPLL algorithm using five real-world datasets: Lost, BirdSong, MSRCv2, Soccer Player, and Yahoo!News. A 2-layer Multi-Layer Perceptron (MLP) model is implemented as the base model. In the pFedPLL, the correlation matrix layer is inserted in the middle to form a three-layer MLP. We observe the same trend as in benchmark datasets, where the pFedPLL achieves the best accuracy with a 0.37–20.67% improvement. Also, FedPLL_LAAR does not perform well. This is because labels in the candidate label set of real-world datasets often have strong correlations (similar labels), making disambiguation harder.

## 5.3 ABLATION STUDY

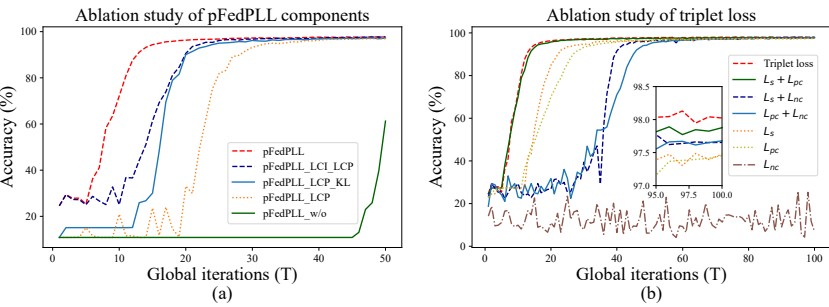

Figure 2: Ablation study: (a) pFedPLL components, (b) triplet loss.

Table 2: Settings for the ablation study of pFedPLL components. All ablations are trained using LeNet on the MNIST dataset.

| Ablation | LCI | LCP | KL score |
|---|---|---|---|
| pFedPLL | ✓ | ✓ | ✓ |
| pFedPLL_LCI_LCP | ✓ | ✓ | |
| pFedPLL_LCP_KL | | ✓ | ✓ |
| pFedPLL_LCP | | ✓ | |
| pFedPLL_w/o | | | |

Table 3: Settings for the ablation study of triplet loss. All ablations are trained using LeNet on the MNIST dataset.

| Ablation | $\mathcal{L}_s$ | $\mathcal{L}_{pc}$ | $\mathcal{L}_{nc}$ |
|---|---|---|---|
| Triplet loss | ✓ | ✓ | ✓ |
| $\mathcal{L}_s + \mathcal{L}_{pc}$ | ✓ | ✓ | |
| $\mathcal{L}_s + \mathcal{L}_{nc}$ | ✓ | | ✓ |
| $\mathcal{L}_{pc} + \mathcal{L}_{nc}$ | | ✓ | ✓ |
| $\mathcal{L}_s$ | ✓ | | |
| $\mathcal{L}_{pc}$ | | ✓ | |
| $\mathcal{L}_{nc}$ | | | ✓ |

**Ablation study of pFedPLL components.** To validate the effectiveness of the components in pFedPLL, we break down the full pFedPLL into four reduced versions by dropping LCI, LCP, and the KL score respectively, as shown in Table 2. Specifically, unchecking LCI removes the twin-module architecture and the correlation matrix layer. Unchecking LCP replaces the triplet loss with the loss function from Fed_LW (which has the second-best performance). Unchecking the KL score uses standard FedAvg aggregation method.

For better presentation, we use $>$ to indicate "is better than". In Figure 2(a), we observe that pFedPLL outperforms all reduced versions of pFedPLL, demonstrating that applying all components in pFedPLL enhances both the accuracy and the convergence speed. ① Comparing LCI, we observe that pFedPLL > pFedPLL_LCP_KL and pFedPLL_LCI_LCP > pFedPLL_LCP. This demonstrates that LCI isolates and protects each worker's unique label correlation, enhancing model performance. ② Comparing LCP, we observe that pFedPLL_LCP > pFedPLL_w/o. This shows that the bi-directional calibration in triplet loss helps effectively distinguish the latent true label. ③ Comparing the KL score, we observe that pFedPLL > pFedPLL_LCI_LCP and pFedPLL_LCP_KL > pFedPLL_w/o, indicating that the KL score helps measure each worker's real contribution, leading to better performance.

**Ablation study of triplet loss.** To validate the effectiveness of the triplet loss in LCP, we evaluate the performance of each individual loss term, all combinations of every two terms, and the complete triplet loss function, as shown in Table 3.

We have the following observations from Figure 2(b). ① The full triplet loss outperforms all variant settings. Each ablation variant shows performance degradation, indicating that every loss term contributes to the effectiveness of pFedPLL. ② The triplet loss $> \mathcal{L}_{pc} + \mathcal{L}_{nc}$, and it also increases rapidly in the early stage of training. Both demonstrate that the summarization term $\mathcal{L}_s$ helps guide the model's update direction by aligning the prediction with the candidate label set. It becomes the cornerstone of the subsequent bi-directional calibration ($\mathcal{L}_{pc}\&\mathcal{L}_{nc}$). ③ $\mathcal{L}_s + \mathcal{L}_{pc} > \mathcal{L}_s$. This demonstrates that $\mathcal{L}_{pc}$ plays a crucial role in effectively identifying the latent true label. $\mathcal{L}_s$ first helps point a correct update direction (falling into the candidate label set) and $\mathcal{L}_{pc}$ then distinguishes between the labelsl in the candidate label set (finding the latent true label). ④. The negative calibration term $\mathcal{L}_{nc}$ acts as a double-edged sword. We observe that $\mathcal{L}_s + \mathcal{L}_{nc}$ and $\mathcal{L}_{pc} + \mathcal{L}_{nc}$ outperform $\mathcal{L}_s$ and $\mathcal{L}_{pc}$ when model converges but underperform during the early stages. This can be explained by the larger $\mathcal{L}_{nc}$ during the initial training phase when the model has not yet identified the correct update direction, causing unstable loss calculations. Once the model finds the correct direction, $\mathcal{L}_{nc}$ enhances the performance. When only applying $\mathcal{L}_{nc}$ alone, the model may fail to converge, because it merely prevents predictions from falling into the non-candidate label set without considering the latent true label.

In summary, every term contributes to the model performance. Implementing the triplet loss ($\mathcal{L}_s + \mathcal{L}_{pc} + \mathcal{L}_{nc}$) can achieve the best performance.

## 5.4 Effect of Candidate Label Set Size

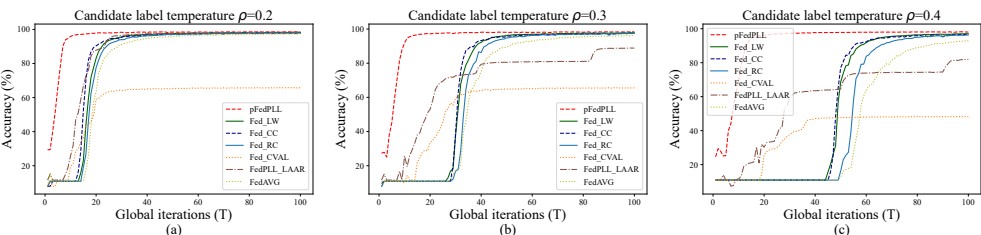

Figure 3: (a)-(c): Accuracy comparison for pFedPLL under different temperature hyperparameters $\rho$ to control the size of the candidate label set: $\rho = 0.2$ (a), $\rho = 0.3$ (b), and $\rho = 0.4$ (c).

Adjusting the temperature hyperparameter $\rho$ controls the candidate label set size. A larger set means a more complicated label correlation, making disambiguation harder for PLL algorithms. In this experiment, we train LeNet on the MNIST dataset and set $\rho = 0.2, 0.3, 0.4$ to obtain average candidate label set sizes of 2.99, 3.97, and 4.93 respectively. The rest of the settings are the same as those used in the benchmark dataset experiment. In Figure 3(a)-(c), we observe that pFedPLL consistently outperforms other algorithms, with a slight performance degradation as $\rho$ increases, while other algorithms degrade quickly. The pFedPLL algorithm surpasses the second-best algorithm by 0.32%, 0.59%, and 1.14% for $\rho = 0.2, 0.3$, and $0.4$, respectively. With the help of the label correlation isolation mechanism and bi-directional calibration loss, pFedPLL first protects each worker's unique label correlation and then accurately identifies the latent true label, both of which are beneficial for handling different levels of candidate label set complexity.

## 6 Conclusion

In this paper, we propose pFedPLL, a personalized federated partial label learning algorithm. We develop label correlation isolation and label correlation personalization to prevent the workers' unique label correlation information from being interfered with while helping identify more accurate learning direction for better performance. We provide a convergence analysis for pFedPLL, demonstrating a convergence rate of $O\left(\sqrt{\frac{1}{T}}\right)$ for smooth non-convex problems. Extensive experiments on both benchmark and real-world datasets illustrate that pFedPLL consistently outperforms SOTA algorithms in a variety of settings. Notably, pFedPLL improved training accuracy by 1.1–49.93% on benchmark datasets and 0.37–17.22% on real-world datasets.

**Reproducibility Statement.** Please refer to Appendix B.3, Table 4 for detailed hyperparameter settings. The source code is available at `https://www.dropbox.com/scl/fo/3abcjortvt8iyf1e9lv7r/AJp2Sf2euxkZPTQNNDAxciw?rlkey=cr4qz2g69d3upyzivu1v99vmm&st=08c1likl&dl=0`.

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

# A  SUPPLEMENTARY MATERIAL: DETAILED CONVERGENCE ANALYSIS

## A.1  PRELIMINARIES

To prove the convergence, we propose a virtual relation module update just like real representation module update (Lines 18–19 in Algorithm 1) as follows:

$$\boldsymbol{w}_{t+1}^{rel} = \sum_{k=1}^{K} s_t^k \boldsymbol{w}_{t,\tau-1}^{k,rel}, \tag{13}$$

$$\boldsymbol{w}_{t+1,0}^{k,rel} = \boldsymbol{w}_{t+1}^{rel}. \tag{14}$$

Thus, we define the complete virtual global model as

$$\boldsymbol{w}_{t+1} \triangleq [\boldsymbol{w}_{t+1}^{rep}, \boldsymbol{w}_{t+1}^{rel}], \tag{15}$$

where $\boldsymbol{w}_{t+1}$ represents the complete global model at the $(t+1)$th global iteration. Based on  equation 14 and Line 19 of Algorithm 1, the local complete model is given as

$$\boldsymbol{w}_{t+1,0}^{k} = [\boldsymbol{w}_{t+1,0}^{k,rep}, \boldsymbol{w}_{t+1,0}^{k,rel}], \forall k \in K. \tag{16}$$

At the start of local training, the initial worker's representation and relation modules are assigned from the global representation and relation modules as described in equation 16. At this point, the local model is identical to the global model. Thus,

$$\boldsymbol{w}_{t+1,0}^{k} = \boldsymbol{w}_{t+1}, \forall k \in K. \tag{17}$$

The gradient of the global loss function can be decomposed into two components, which we define as

$$\nabla F(\boldsymbol{w}_{t+1}) \triangleq [\nabla_{\boldsymbol{w}^{rep}} F(\boldsymbol{w}_{t+1}), \nabla_{\boldsymbol{w}^{rel}} F(\boldsymbol{w}_{t+1})]. \tag{18}$$

The gradient of the worker's loss function can be decomposed into two components, which we define as

$$\nabla F_k(\boldsymbol{w}_{t+1,h}^{k}) \triangleq [\nabla_{\boldsymbol{w}^{rep}} F_k(\boldsymbol{w}_{t+1,h}^{k})), \nabla_{\boldsymbol{w}^{rel}} F_k(\boldsymbol{w}_{t+1,h}^{k})], \forall k \in K, \forall h \in [0, \dots, \tau-1]. \tag{19}$$

Based on the definition in equation 19, we have

$$\begin{aligned}
\|\nabla F_k(\boldsymbol{w}_{t+1,h}^{k})\|_2^2 &= \|[\nabla_{\boldsymbol{w}^{rep}} F_k(\boldsymbol{w}_{t+1,h}^{k})), \nabla_{\boldsymbol{w}^{rel}} F_k(\boldsymbol{w}_{t+1,h}^{k})]\|_2^2 \\
&= (\|[\nabla_{\boldsymbol{w}^{rep}} F_k(\boldsymbol{w}_{t+1,h}^{k})), 0] + [0, \nabla_{\boldsymbol{w}^{rel}} F_k(\boldsymbol{w}_{t+1,h}^{k})]\|_2)^2 \\
&= \|\nabla_{\boldsymbol{w}^{rep}} F_k(\boldsymbol{w}_{t+1,h}^{k})\|_2^2 + 2\langle [\nabla_{\boldsymbol{w}^{rep}} F_k(\boldsymbol{w}_{t+1,h}^{k})), 0], [0, \nabla_{\boldsymbol{w}^{rel}} F_k(\boldsymbol{w}_{t+1,h}^{k})] \rangle \\
&\quad + \|\nabla_{\boldsymbol{w}^{rel}} F_k(\boldsymbol{w}_{t+1,h}^{k})\|_2^2. \\
&= \|\nabla_{\boldsymbol{w}^{rep}} F_k(\boldsymbol{w}_{t+1,h}^{k})\|_2^2 + \|\nabla_{\boldsymbol{w}^{rel}} F_k(\boldsymbol{w}_{t+1,h}^{k})]\|_2^2, \forall k \in K, \forall h \in [0, \dots, \tau-1]
\end{aligned} \tag{20}$$

According to Lines 8–9 in Algorithm 1, equation 16 and equation 19, we have the local update rule as

$$\boldsymbol{w}_{t,h+1}^{k} = \boldsymbol{w}_{t,h}^{k} - \eta \nabla F_k(\boldsymbol{w}_{t,h}^{k}). \tag{21}$$

To calculate the difference between $h=0$ and $h=\tau-1$ according to equation 21, we have

$$\boldsymbol{w}_{t,0}^{k} - \boldsymbol{w}_{t,\tau-1}^{k} = \eta \sum_{h=0}^{\tau-1} \nabla F_k(\boldsymbol{w}_{t,h}^{k}). \tag{22}$$

According to equation 17, equation 19, and rearranging equation 22, we obtain

$$\boldsymbol{w}_t - \boldsymbol{w}_{t,\tau-1}^{k} = \eta \sum_{h=0}^{\tau-1} [\nabla_{\boldsymbol{w}^{rep}} F_k(\boldsymbol{w}_{t,h}^{k}), \nabla_{\boldsymbol{w}^{rel}} F_k(\boldsymbol{w}_{t,h}^{k})]. \tag{23}$$

Based on Lines19 in Algorithm 1, equation 13, and equation 15, we have

$$\boldsymbol{w}_t - \boldsymbol{w}_{t+1} = \boldsymbol{w}_t - \sum_{k=1}^{K} s_t^k(\boldsymbol{w}_{t,\tau-1}^k) = \sum_{k=1}^{K} s_t^k(\boldsymbol{w}_t - \boldsymbol{w}_{t,\tau-1}^k) \tag{24}$$

Substituting the equation 23 into equation 24, we have

$$\boldsymbol{w}_t - \boldsymbol{w}_{t+1} = \eta \sum_{k=1}^{K} s_t^k \sum_{h=0}^{\tau-1} [\nabla_{\boldsymbol{w}^{rep}} F_k(\boldsymbol{w}_{t,h}^k), \nabla_{\boldsymbol{w}^{rel}} F_k(\boldsymbol{w}_{t,h}^k)] \tag{25}$$

By rearranging equation 25, we derive the global update rule as

$$\boldsymbol{w}_{t+1} = \boldsymbol{w}_t - \eta \sum_{k=1}^{K} s_t^k \sum_{h=0}^{\tau-1} [\nabla_{\boldsymbol{w}^{rep}} F_k(\boldsymbol{w}_{t,h}^k), \nabla_{\boldsymbol{w}^{rel}} F_k(\boldsymbol{w}_{t,h}^k)]. \tag{26}$$

We assume $F_k(\cdot)$ satisfies the following standard conditions that are necessary for theoretical analysis (Wang et al., 2019; Yang et al., 2022; Huo et al., 2020).

**Assumption 1.** *(Bounded diversity). The variance of stochastic gradient on local workers is upper bounded. So that any $k \in \{1, \ldots, K\}$, it is satisfied that*

$$\mathbb{E}_{\xi \sim D_k} \|\nabla F_k(\boldsymbol{w}, \xi) - \nabla F_k(\boldsymbol{w})\|_2^2 \leq \delta^2, \forall \boldsymbol{w}, k.$$

This is equivalent to

$$\mathbb{E}_{\xi \sim D_k} [\|\nabla_{\boldsymbol{w}_{rel}} F_k(\boldsymbol{w}, \xi) - \nabla_{\boldsymbol{w}_{rel}} F_k(\boldsymbol{w})\|_2^2 + \|\nabla_{\boldsymbol{w}_{rep}} F_k(\boldsymbol{w}, \xi) - \nabla_{\boldsymbol{w}_{rep}} F_k(\boldsymbol{w})\|_2^2] \leq \delta^2, \forall \boldsymbol{w}, k.$$

**Assumption 2.** *(L-Lipschitz). The gradients of $F_k$ and $F$ are Lipschitz continuous with a constant $L > 0$, so that any $k \in \{1, \ldots, K\}$, it is satisfied that*

$$\|\nabla F_k(\boldsymbol{w}_1) - \nabla F_k(\boldsymbol{w}_2)\|_2 \leq L\|\boldsymbol{w}_1 - \boldsymbol{w}_2\|_2, \forall \boldsymbol{w}_1, \boldsymbol{w}_2, k,$$

$$\|\nabla F(\boldsymbol{w}_1) - \nabla F(\boldsymbol{w}_2)\|_2 \leq L\|\boldsymbol{w}_1 - \boldsymbol{w}_2\|_2, \forall \boldsymbol{w}_1, \boldsymbol{w}_2.$$

A.2 CONVERGENCE ANALYSIS

In Lemma 1, we first prove the upper bound of pFedPLL between $F(\boldsymbol{w}_{t+1})$ and $F(\boldsymbol{w}_t)$.

**Lemma 1.** *Under Assumptions 1 and 2, the update of $\boldsymbol{w}_t$ on the server at each global aggregation is upper bounded as*

$$\mathbb{E}_{\xi,k} F(\boldsymbol{w}_{t+1}) \leq F(\boldsymbol{w}_t) - \frac{\eta}{2}\tau \sum_{k=1}^{k} s_t^k \mathbb{E}_{\xi,k} \|\nabla F_k(\boldsymbol{w}_t)\|_2^2 + (\frac{L\tau\eta + 1}{2})\eta^2 L\tau\delta^2$$

$$- (\frac{\eta}{2} - \frac{\eta}{2}L^2\eta^2\tau^2 - \frac{L}{2}\eta^2\tau) \sum_{k=1}^{k} s_t^k \sum_{h=0}^{\tau-1} \mathbb{E}_{\xi,k} \|\nabla F_k(\boldsymbol{w}_{t,h}^k)\|_2^2.$$

*Proof.* According to Assumptions 2 and equation 18, it holds that

$$\mathbb{E} F(\boldsymbol{w}_{t+1}) \leq F(\boldsymbol{w}_t) + \mathbb{E}\langle \nabla_{\boldsymbol{w}^{rel}} F(\boldsymbol{w}_t), \boldsymbol{w}_{t+1}^{rel} - \boldsymbol{w}_t^{rel}\rangle$$

$$+ \mathbb{E}\langle \nabla_{\boldsymbol{w}^{rel}} F(\boldsymbol{w}_t), \boldsymbol{w}_{t+1}^{rel} - \boldsymbol{w}_t^{rel}\rangle$$

$$+ \frac{L}{2}\mathbb{E}\|\boldsymbol{w}_{t+1}^{rel} - \boldsymbol{w}_t^{rel}\|_2^2 + \frac{L}{2}\mathbb{E}\|\boldsymbol{w}_{t+1}^{rep} - \boldsymbol{w}_t^{rep}\|_2^2. \tag{27}$$

By taking the expectation over the samples, rearranging the inequality in equation 27, and considering equation 25 and equation 26, we obtain

$$\mathbb{E}_{\xi} F(\boldsymbol{w}_{t+1}) \leq F(\boldsymbol{w}_t) - \langle \nabla_{\boldsymbol{w}^{rep}} F(\boldsymbol{w}_t), \mu \sum_{k=1}^{K} s_t^k \sum_{h=0}^{\tau-1} \nabla_{\boldsymbol{w}^{rep}} F_k(\boldsymbol{w}_{t,h}^k, \xi)\rangle$$

$$+ \frac{L}{2} \mathbb{E}_\xi \| \eta \sum_{k=1}^{K} s_t^k \sum_{h=0}^{\tau-1} \nabla_{\boldsymbol{w}^{rep}} F_k(\boldsymbol{w}_{t,h}^k, \xi) \rangle \|_2^2$$

$$- \langle \nabla_{\boldsymbol{w}^{rel}} F(\boldsymbol{w}_t), \mu \sum_{k=1}^{K} s_t^k \sum_{h=0}^{\tau-1} \nabla_{\boldsymbol{w}^{rel}} F_k(\boldsymbol{w}_{t,h}^k, \xi) \rangle$$

$$+ \frac{L}{2} \mathbb{E}_\xi \| \eta \sum_{k=1}^{K} s_t^k \sum_{h=0}^{\tau-1} \nabla_{\boldsymbol{w}^{rel}} F_k(\boldsymbol{w}_{t,h}^k, \xi) \rangle \|_2^2, \tag{28}$$

where the inequality follows from $\mathbb{E}_\xi[\nabla_{\boldsymbol{w}^{rep}} F_k(\boldsymbol{w}, \xi)] = \nabla_{\boldsymbol{w}^{rep}} F_k(\boldsymbol{w})$ and $\mathbb{E}_\xi[\nabla_{\boldsymbol{w}^{rel}} F_k(\boldsymbol{w}, \xi)] = \nabla_{\boldsymbol{w}^{rel}} F_k(\boldsymbol{w})$. Taking the expectations over the workers, we have

$$\mathbb{E}_{\xi,k} F(\boldsymbol{w}_{t+1}) \leq F(\boldsymbol{w}_t) - \frac{\eta}{2} \sum_{k=1}^{k} s_t^k \sum_{h=0}^{\tau-1} \mathbb{E}_{\xi,k}[\|\nabla F_k(\boldsymbol{w}_t)\|_2^2]$$

$$- \frac{\eta}{2} \sum_{k=1}^{k} s_t^k \sum_{h=0}^{\tau-1} \mathbb{E}_{\xi,k}[\|\nabla F_k(\boldsymbol{w}_{t,h}^k)\|_2^2]$$

$$+ \frac{\eta}{2} \sum_{k=1}^{k} s_t^k \sum_{h=0}^{\tau-1} \underbrace{\mathbb{E}_{\xi,k}[\|\nabla_{\boldsymbol{w}^{rep}} F_k(\boldsymbol{w}_t) - \nabla_{\boldsymbol{w}^{rep}} F_k(\boldsymbol{w}_{t,h}^k))\|_2^2 + \|\nabla_{\boldsymbol{w}^{rel}} F_k(\boldsymbol{w}_t) - \nabla_{\boldsymbol{w}^{rel}} F_k(\boldsymbol{w}_{t,h}^k))\|_2^2]}_{Q1}$$

$$+ \frac{L}{2} \eta^2 \sum_{k=1}^{k} s_t^k \mathbb{E}_{\xi,k}[\| \underbrace{\sum_{h=0}^{\tau-1} \nabla F_k(\boldsymbol{w}_{t,h}^k, \xi) \rangle}_{Q2} \|_2^2, \tag{29}$$

where the inequality follows from Jensen's inequality, $\langle a, b \rangle = \frac{1}{2}(\|a\|_2^2 + \|b\|_2^2 - \|a-b\|_2^2)$. We next prove the upper bound of Q1 as

$$Q1 \leq L^2 \eta^2 \mathbb{E}_{\xi,k} \| \sum_{j=0}^{h-1} (\nabla_{\boldsymbol{w}^{rep}} F_k(\boldsymbol{w}_{t,j}^k, \xi) - \nabla_{\boldsymbol{w}^{rep}} F_k(\boldsymbol{w}_{t,j}^k))\|_2^2$$

$$+ L^2 \eta^2 \mathbb{E}_{\xi,k} \| \sum_{j=0}^{h-1} (\nabla_{\boldsymbol{w}^{rel}} F_k(\boldsymbol{w}_{t,j}^k, \xi) - \nabla_{\boldsymbol{w}^{rel}} F_k(\boldsymbol{w}_{t,j}^k))\|_2^2$$

$$+ L^2 \eta^2 \mathbb{E}_{\xi,k}(\| (\sum_{j=0}^{h-1} \nabla_{\boldsymbol{w}^{rep}} F_k(\boldsymbol{w}_{t,j}^k))\|_2^2 + \| (\sum_{j=0}^{h-1} \nabla_{\boldsymbol{w}^{rel}} F_k(\boldsymbol{w}_{t,j}^k))\|_2^2)$$

$$\leq L^2 \eta^2 \delta^2 h + L^2 \eta^2 h \sum_{j=0}^{h-1} \mathbb{E}_{\xi,k} \|\nabla F_k(\boldsymbol{w}_{t,j}^k)\|_2^2, \tag{30}$$

where the first inequality follows from Assumption 1, $\boldsymbol{w}_t = \boldsymbol{w}_{t,0}^k$, and $\mathbb{E}\|z_1 + \ldots + z_n\|_2^2 \leq \mathbb{E}[\|z_1\|_2^2 + \ldots + \|z_n\|_2^2]$ for any $z_1, \ldots, z_2$. The last inequality is from the Assumption 2, and equation 20. It sums the gradient difference from $j = 0$ to $j = h - 1$. Since the maximum value of $h$ is $\tau$, we replace $h$ with $\tau$ in equation 30 and sum from $h = 0$ to $h = \tau - 1$. This still satisfies the inequality in equation 30. Then, we have

$$Q1 \leq L^2 \eta^2 \delta^2 \tau + L^2 \eta^2 \tau \sum_{h=0}^{\tau-1} \mathbb{E}_{\xi,k} \|\nabla F_k(\boldsymbol{w}_{t,h}^k)\|_2^2, \tag{31}$$

Summing the inequality in equation 31 from $h = 0$ to $\tau - 1$, we have

$$\sum_{h=0}^{\tau-1} Q1 \leq L^2 \eta^2 \delta^2 \tau^2 + L^2 \eta^2 \tau^2 \sum_{h=0}^{\tau-1} \mathbb{E}_{\xi,k} \|\nabla F_k(\boldsymbol{w}_{t,h}^k)\|_2^2, \tag{32}$$

where the inequality comes from $h \leq \tau - 1$. Then we going to prove the upper bound of Q2 as

$$Q2 = \mathbb{E}_{\xi,k} \| \sum_{h=0}^{\tau-1} \nabla F_k(\boldsymbol{w}_{t,h}^k, \xi) - \nabla F_k(\boldsymbol{w}_{t,h}^k) + \nabla F_k(\boldsymbol{w}_{t,h}^k) \|_2^2 \leq \delta^2 \tau + \tau \sum_{h=0}^{\tau-1} \mathbb{E}_{\xi,k} \| \nabla F_k(\boldsymbol{w}_{t,h}^k) \|_2^2,$$

$$\tag{33}$$

where the inequality comes from Assumption 1. Substituting the upper bound of Q1 and Q2 into inequality equation 29, we have

$$\mathbb{E}_{\xi,k} F(\boldsymbol{w}_{t+1}) \leq F(\boldsymbol{w}_t) - \frac{\eta}{2} \tau \sum_{k=1}^{k} s_t^k \mathbb{E}_{\xi,k} \| \nabla F_k(\boldsymbol{w}_t) \|_2^2 + (\frac{L\tau\eta + 1}{2}) \eta^2 L \tau \delta^2$$

$$- (\frac{\eta}{2} - \frac{\eta}{2} L^2 \eta^2 \tau^2 - \frac{L}{2} \eta^2 \tau) \sum_{k=1}^{k} s_t^k \sum_{h=0}^{\tau-1} \mathbb{E}_{\xi,k} \| \nabla F_k(\boldsymbol{w}_{t,h}^k) \|_2^2. \tag{34}$$

We have completed the proof of Lemma 1. $\qquad\square$

Based on Lemma 1, we can prove the convergence of pFedPLL by telescopically summing from $t = 0$ to $t = T - 1$.

**Theorem 2.** *Suppose (1) $\eta \leq \frac{1}{2L\tau}, \forall t \in 0, \ldots, T-1$, and (2) $\exists F_{inf}$ is the lower bound of $F(\cdot)$, we have*

$$\min_{t \in \{0,\ldots,T-1\}} \mathbb{E}_{\xi,k} \| \nabla F(\boldsymbol{w}_t) \|_2^2 \leq \frac{4L}{T} (F(\boldsymbol{w}_0) - F_{inf})) + 3\eta^2 L^2 \delta^2.$$

*Proof.* According to Lemma 1, by setting $\eta \leq \frac{1}{2L\tau}$, we ensure that $\frac{\eta}{2} - \frac{\eta}{2} L^2 \eta^2 \tau^2 - \frac{L}{2} \eta^2 \tau \geq 0$. This condition allows us to prove that the algorithm is guaranteed to converge to critical points for smooth non-convex problems. Considering the above condition and taking the expectation of inequality equation 34, upon rearranging, it holds that

$$\mathbb{E}_{\xi,k} \| \nabla F(\boldsymbol{w}_t) \|_2^2 \leq 4L (\mathbb{E}_{\xi,k} F(\boldsymbol{w}_t) - \mathbb{E}_{\xi,k} F(\boldsymbol{w}_{t+1})) + 3\eta^2 L^2 \delta^2. \tag{35}$$

Summing it from $t = 0$ to $T - 1$ we have

$$\sum_{t=0}^{T-1} \mathbb{E}_{\xi,k} \| \nabla F(\boldsymbol{w}_t) \|_2^2 \leq 4L (F(\boldsymbol{w}_0) - F(\boldsymbol{w}_T)) + 3\eta^2 L^2 \delta^2 T. \tag{36}$$

Let $F_{inf} \leq F(\boldsymbol{w}_T)$, we have that

$$\min_{t \in \{0,\ldots,T-1\}} \mathbb{E}_{\xi,k} \| \nabla F(\boldsymbol{w}_t) \|_2^2 \leq \frac{4L}{T} (F(\boldsymbol{w}_0) - F_{inf})) + 3\eta^2 L^2 \delta^2.$$

We have completed the proof of the Theorem 2. $\qquad\square$

Under the condition $\eta \leq \frac{1}{2L\tau}$, we have proven that pFedPLL is guaranteed to converge to critical points for smooth non-convex problems at a rate of $O\left(\sqrt{\frac{1}{T}}\right)$.

# B SUPPLEMENTARY MATERIAL: ADDITIONAL EXPERIMENT SETUP AND RESULT

## B.1 EXPERIMENT DATASET

**Benchmark datasets.** We utilize MNIST (LeCun et al., 1998), Fashion-MNIST (F-MNIST) (Xiao et al., 2017), Kuzushiji-MNIST (K-MNIST) (Clanuwat et al., 2018), and CIFAR-10 (Krizhevsky et al., 2009). The MNIST, F-MNIST, and K-MNIST datasets, each consists of 60,000 training images and 10,000 test images. Each image is a grayscale $28 \times 28$ pixel grid categorized into 10 classes. MNIST, F-MNIST, and K-MNIST are designed for handwritten digit classification, fashion item classification, and Japanese character classification, respectively. The CIFAR-10 dataset

includes 50,000 training images and 10,000 test images, with each $32 \times 32 \times 3$ colored image in RGB format organized into 10 classes for object classification tasks.

**Real-world partial label datasets.** We use five real-world partial label datasets, including Lost (Cour et al., 2011), BirdSong (Briggs et al., 2012), MSRCv2 (Liu & Dietterich, 2012), Soccer Player (Zeng et al., 2013), and Yahoo!News (Guillaumin et al., 2010). The Lost dataset, based on the TV series "Lost", includes 1,122 face images with 108 features, categorized into 16 names with an average candidate label set size of 2.23. The Birdsong dataset focuses on bird song classification, featuring 4,998 data points and 38 features, grouped into 13 categories with an average candidate label set size of 2.18. The MSRCv2 dataset, aimed at object classification, contains 1,758 data points with 48 features, categorized into 23 objects and an average candidate label set size of 3.16. The Soccer Player dataset includes 17,472 data points with 279 features for naming soccer players, categorized into 171 names, and an average candidate label set size of 2.09. Lastly, the Yahoo!News dataset for automatic face naming in news articles includes 22,991 data points with 163 features, grouped into 219 names, with an average candidate label set size of 1.91.

### B.2 THE PARTIAL LABEL DATASET GENERATION

Since the benchmark datasets are primarily for supervised learning, we follow the instance-dependent generation process (Xu et al., 2021) to manually corrupt them into partial label datasets. Specifically, we determine the flipping probability for each incorrect label corresponding to an instance $\boldsymbol{x}_i$ using the confidence predictions from a clean neural network, $\hat{\theta}$, trained on the original supervised dataset. The flip probability for each incorrect label for instance $\boldsymbol{x}_i$ is calculated as:

$$\zeta_j = \frac{f_j(\boldsymbol{x}_i; \hat{\theta})}{\max_{z \in \tilde{\boldsymbol{Y}}_i} f_z(\boldsymbol{x}_i; \hat{\theta})} \rho, \forall j \in \tilde{\boldsymbol{Y}}_i, \tag{37}$$

where $\tilde{\boldsymbol{Y}}_i$ is the set of all incorrect labels except for the true label of $\boldsymbol{x}_i$, and $f_j(\boldsymbol{x}_i; \hat{\theta})$ is the confidence prediction of the clean neural network $\hat{\theta}$. $\rho \in [0, 1]$ is the temperature hyperparameter to control the candidate label set size.

### B.3 EXPERIMENT HYPERPARAMETER SETTINGS

Table 4: Experiment hyperparameter settings.

| Experiments | Hyperparameters | | | | | | |
|---|---|---|---|---|---|---|---|
| | $T$ | $\tau$ | $\eta$ | $\beta$ | $\rho$ | $K$ | Batch size |
| Performance comparison (Table 1) | 100 | 40 | 0.01 | 0.5 | 0.4 | 4 | 256 (Benchmark dataset) 32 (Real-world partial label dataset) |
| Ablation study of pFedPLL components (Fig. 2(a)) | 100 | 40 | 0.01 | 0.5 | 0.4 | 4 | 256 |
| Ablation study of triplet loss (Fig. 2(b)) | 100 | 40 | 0.01 | 0.5 | 0.4 | 4 | 256 |
| Effect of Candidate Label Set Size (Fig. 3(a)–(c)) | 100 | 40 | 0.01 | 0.5 | 0.2,0.3,0.4 | 4 | 256 |
| Effects of number of workers (Fig. 4) | 100 | 40 | 0.01 | 0.5 | 0.4 | 20,40,80 | 256 |

### B.4 EFFECT OF NUMBER OF WORKERS

To demonstrate the impact of varying the number of workers on the performance of pFedPLL methods, we set the number of workers, $K$, to 20, 40, and 80. This experiment employs LeNet on the MNIST dataset. In Figure 4, the results show that as the number of workers increases, the convergence performance of pFedPLL degrades, aligning with our expectations, i.e., as the number of workers grows, the data variance among workers increases, leading to decreased convergence performance. Notably, even with up to 80 workers, pFedPLL maintains reasonable convergence performance. In summary, the results demonstrate that pFedPLL effectively handles PLL tasks in an FL setting, maintaining performance even with an increased number of workers.

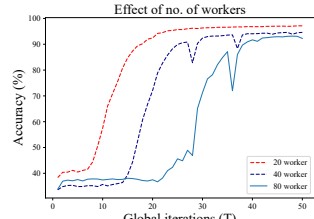

Figure 4: Effect of numbers of workers in pFedPLL.

