# OpenReview forum: "PERSONALIZED FEDERATED PARTIAL LABEL LEARNING"
_ICLR.cc/2025/Conference — ICLR 2025 Conference Withdrawn Submission_

### Official Review · Reviewer_JhMp · 2024-10-25

**Soundness:** 1
**Presentation:** 1
**Contribution:** 1
**Rating:** 3
**Confidence:** 4

**Summary:**

Partial label learning is an important learning technique on the datasets with inaccurate labels. Existing work focus on the centralized setting. This paper studies partial label learning in federated learning scenarios. This paper provides theoretical analysis as well as experiments.

**Strengths:**

The problem has some (limited) novelty.

**Weaknesses:**

1. The problem is not interesting. Can the authors provide some examples that needs PLL in FL?

2. It is not clear what is the additional challenge of partial label learning in FL. What will happen if we directly combine partial label learning and FL methods?

3. In general, the paper is hard to follow.

4. For the theoretical result, it is necessary to make a comparison with the ordinary partial label learning, and FL with full  labels. Moreover, assumptions should be stated explicitly in the main paper instead of in the appendix.

5. In section 5.1, at the bottom of page 7, the authors need to cite the original paper of Fed_CC, Fed_RC, Fed_CVAL, Fed_LW.

6. A minor issue. It would be better to change the term "Comparison methods" to "Baseline methods" in section 5.1.

7. Why does Fed_CVAL performs significantly worse than FedAvg?

8. In general, the paper does not put forward any interesting point. I do not see any novelty that make this paper different from existing PLL paper or existing FL paper.

9. What is $\tau$ in eq.(13)?

**Questions:**

See weaknesses above.

---

### Official Review · Reviewer_f5Wh · 2024-10-29

**Soundness:** 2
**Presentation:** 2
**Contribution:** 2
**Rating:** 3
**Confidence:** 4

**Summary:**

This paper proposed a personalized federated learning algorithm for the label correlation interference problem
in federated partial label learning with non-i.i.d. data among clients,
and proved that the algorithm achieves a convergence rate of $O(1/\sqrt{T})$ for smooth non-convex loss functions.
The proposed algorithm contains a new twin-module architecture which keeps the label correlation information local
and only shares the representations with others,
and a new bi-directional calibration loss which aligns the prediction result with the true label
and pushes away the prediction result that falls into the non-candidate label set.
Finally, numerical experiments were conducted to validate the effiency of the proposed algorithm.

**Strengths:**

This paper invents effective techniques for federated partial label learning and centralized partial label learning.
The experiments are also sufficient.

**Weaknesses:**

Although the experiments have verified the effectiveness of the proposed algorithm,
the theoretical contribution seems weak.

(1) Theorem 1 can not demonstrate that federated learning improves the convergence rate.
In other words, the convergence rate is independent of the number of clients, $K$,
and is same with that of centralized learning.

(2) I also doubt the correctness of Theorem 1.
The authors claimed that the convergence rate is $O(1/\sqrt{T})$.
However, by Theorem 1, if we set $\tau =\sqrt{T}$, then the convergence rate is $O(1/T)$.
What's more, it seems that the smaller the value of $\eta$ is, the better the convergence will be.
However, such a convergence rate i.e., $O(1/T)$, contradicts a konwn lower bound [1].
For smooth non-convex loss functions, the convergence rate of any stochastic first-order algorithms must be $\Omega(1/\sqrt{T})$[1].


References

[1] Arjevani et al. Lower bounds for non-convex stochastic optimization. Mathematical Programming, 2023.

**Questions:**

(1) Could the proposed algorithm be used to the case of i.i.d. data?
It seems that the proposed algorithm has assumed that the data is non-i.i.d.
However, it is unknown to us that whether the data is non-i.i.d.

(2) In Section 3.1, the definition of $M^k_i$ is somewhat unclear to me.
To be specific, $y^k_{i,j}\in \{0,1}$, while the elements in $Y^k_i$ should be a vector with lengeth $C$.
So, the notation $y^k_{i,j} \in Y^k_i$ is unclear to me.

---

### Official Review · Reviewer_a8X3 · 2024-11-02

**Soundness:** 2
**Presentation:** 2
**Contribution:** 2
**Rating:** 3
**Confidence:** 3

**Summary:**

The paper proposes pFedPLL (personalized federated partial label learning) to address label correlation interference in federated learning (FL) with non-i.i.d. partial label datasets. The method introduces two core mechanisms:

- Label Correlation Isolation (LCI): It isolates each worker’s label correlation matrix to prevent interference during model aggregation.
- Label Correlation Personalization (LCP): It incorporates a bi-directional calibration loss to guide learning toward the true label within the candidate label set and away from non-candidate labels.

**Strengths:**

1. The paper addresses a critical issue in federated partial label learning — the label correlation interference caused by data heterogeneity. The twin-module architecture with LCI is a unique approach, and LCP’s bi-directional calibration is an innovative way to leverage partial label data.

2. The method is well-structured with clear explanations of the proposed mechanisms, algorithms, and theoretical insights. Visual aids effectively illustrate the twin-module architecture and triplet loss setup.

**Weaknesses:**

1. The novelty of the proposed method is not clear. For example, sharing representation layers and constructing a correlation matrix and the final linear layer are standard methods. The novelty of the proposed positive calibration and negative calibration is also not clear.
2. The optimality of using the proposed KL scores is not clear.
3. Although pFedPLL performs well on relatively small datasets, the paper does not discuss potential scalability issues when dealing with significantly larger models or a larger number of clients. Expanding on computational and memory overhead in highly distributed settings would strengthen the practical applicability.

**Questions:**

Please address the weakness above.

---

### Official Review · Reviewer_mPeu · 2024-11-05

**Soundness:** 2
**Presentation:** 2
**Contribution:** 2
**Rating:** 3
**Confidence:** 4

**Summary:**

This paper proposed a personalized FL to address the label correlation interference problem in partial label learning.

**Strengths:**

This paper addresses a weakly supervised problem in federated learning where each client has access only to a candidate label set for each instance. This problem formulation aligns more closely with real-world FL applications, potentially reducing the burden of collecting
 a large amount of high quality data instances.

**Weaknesses:**

1. The problem of partial labeling in federated learning is indeed interesting. However, the motivation for introducing label correlation interference in heterogeneous FL settings is somewhat unclear. In particular, it's not immediately evident why instances on different clients might be labeled with varying candidate label sets. To strengthen the rationale behind this assumption, it would be helpful for the authors to provide concrete examples from real-world applications where such label correlation interference is likely to occur.

2. The proposed solution seems not novel to me. As the parameter personalization has been widely explored in traditional FL research such as FedRep, FedPer, and PartialFed. More specifically, I do not see much differences between the proposed  method and FedPer or FedRep, as the label correlation matrix layer is a linear layer that can be directly merged into the classifier.

3. Equations (6) and (7) appear somewhat unusual. It seems to me that \alpha is directly proportional to \wave{y}, so what is the difference between Eq. 7 and the traditional cross-entropy loss?

4. What is the relationship between the convergence analysis and partially labeling? How does the partially labeling affect the convergence speed?

5. The datasets used in the experiments are overly simple; the authors should consider evaluating their approach on larger-scale datasets, such as Tiny-ImageNet, to better demonstrate its effectiveness.

6. In the experiments, the method for creating datasets with varying label correlations across clients is unclear. It would be helpful for the authors to clarify how they generated datasets with different label correlation structures.

7. In Fig. 2 (b)， why does the training using only the negative  calibration unconverged but plus the other losses can make it converge.

[1] Federated learning with personalization layers, 2019.
[2] Exploiting shared representations for personalized federated learning, ICML 2021.
[3] Partialfed: Cross-domain personalized federated learning via partial initialization, Neurips 2021.

**Questions:**

Please refer to the weakness section.

---

### Note · Authors · 2024-11-25

I have read and agree with the venue's withdrawal policy on behalf of myself and my co-authors.